# Role of Testing Conditions in Formation of Tribological Layers at Line Contacts of Antifriction CF-Reinforced PI- and PEI-Based Composites [note 1]

**DOI:** 10.3390/molecules27196376

**Published:** 2022-09-27

**Authors:** Sergey V. Panin, Jiangkun Luo, Dmitry G. Buslovich, Vladislav O. Alexenko, Lyudmila A. Kornienko, Anton V. Byakov, Vitaly N. Paimushin, Artur R. Shugurov

**Affiliations:** 1Laboratory of Mechanics of Polymer Composite Materials, Institute of Strength Physics and Materials Science of Siberian Branch of Russian Academy of Sciences, 634055 Tomsk, Russia; 2Department of Materials Science, Engineering School of Advanced Manufacturing Technologies, National Research Tomsk Polytechnic University, 634050 Tomsk, Russia; 3Kazan Scientific Center of Russian Academy of Sciences, 420111 Kazan, Russia

**Keywords:** polyimide, polyetherimide, wear rate, carbon fiber, polytetrafluoroethylene, molybdenum disulfide, nanoindentation, friction coefficient, tribological layer, line contact

## Abstract

High-strength PI and PEI polymers differ by chemical structure and flexibility of the polymer chains that ensure lower cost and higher manufacturability of the latter. The choice of a particular polymer matrix is of actuality at design of antifriction composites on their basis. In this study, a comparative analysis of tribological behavior of PI and PEI- based composites was carried out with linear contact rubbing. The neat materials, as well as the two- and three-component composites reinforced with chopped carbon fibers, were investigated. The third components were typically used, but were different in nature (polymeric and crystalline) being solid lubricant fillers (PTFE, graphite and MoS_2_) with characteristic dimensions of several microns. The variable parameters were both load and sliding speed, as well as the counterface material. It was shown that an improvement of the tribological properties could be achieved by the tribological layer formation, which protected their wear track surfaces from the cutting and plowing effects of asperities on the surfaces of the metal and ceramic counterparts. The tribological layers were not formed in both neat polymers, while disperse hardening by fractured CF was responsible for the tribological layer formation in both two- and three component PI- and PEI-based composites. The effect of polymer matrix in tribological behavior was mostly evident in two-component composites (PI/CF, PEI/CF) over the entire *P*⋅*V* product range, while extra loading with Gr and MoS_2_ leveled the regularities of tribological layer formation, as well as the time variation in friction coefficients.

## 1. Introduction

Thermoplastic polyimide (PI) and polyetherimide (PEI) are a class of high-performance polymers (HPP) that have been widely employed in high-technology fields, such as aerospace, marine, mechanical, and automotive industries, as a result of their high strength-to-weight ratio, superior thermal stability and mechanical properties [1,2,3,4]. The design of thermoplastic polyetherimide (PEI) is obtained on the basis of polyimide by loading repeating units, consisting of aromatic and heterocyclic fragments, with ‘hinged’ oxygen atoms [5]. Despite the increase in chain flexibility and the decrease in the glass transition temperature, a number of basic PI characteristics are retained, including advanced physical and mechanical properties, as well as the decomposition onset temperature [6,7]. Nevertheless, the pristine PI and PEI have the demerits of poor wear-resistance, which severely limit their utility as tribological components in harsh operating conditions, especially under high *P*⋅*V* (the product of normal pressure and linear velocity) conditions. [8,9,10]. Research has shown that these properties can be attained by fabricating polymer composites through adding fibers and solid lubricants (polytetrafluoroethylene (PTFE), graphite (Gr), molybdenum disulfide (MoS_2_), etc.). The fibers bear most of the load and diminish the wear rate by reducing the subsurface deformation and interrupting crack propagation [11,12,13]. In addition, solid lubricants usually reduce the friction coefficient; and furthermore, would commonly promote a homogeneous transfer film on the counterpart surface. The latter reduces the wear rate by avoiding direct contact with asperities on the counterpart surface [14,15,16].

The interactions between the fiber ends and counterpart can induce the high stress concentration and temperature flashes at the sliding interfaces as reported by Chang and Friedrich [17], and thereby lead to the occurrence of complex physicochemical interactions between polymer matrix and counterpart. In the meantime, the wear resistance of neat PEI, especially at high temperature and under high *P*⋅*V*-factor conditions, was remarkably improved by the addition of SCFs in a range from 5 to 20 vol % [18]. The topography of the worn surfaces indicates that the incorporation of SCFs helped the formation of transfer layers on both the counterface (transfer film) and worn surface of the PEI composites (tribofilm). In doing so, the transfer film became more continuous with increasing test temperature [18].

Song et al. investigated the effects of SCF-EG hybrid on the mechanical, tribological and thermal performances of PEI composites. The results indicated that the strength and fracture toughness of PEI composites are enhanced by replacing expanded graphite (EG) with Short Carbon Fibers (SCFs). The simultaneous addition of SCFs and EG shows synergistic effects on the Young’s modulus, flexural modulus, friction coefficient and specific wear rate of PEI composites [19]. Song et al. investigated tribological behaviors and *P*⋅*V* limit of chopped carbon fiber, glass fibers and MoS_2_ reinforced PTFE composites. The experimental results revealed the best tribological properties with the highest *P*⋅*V* limits of 9.5 MPa m/s at 1 m/s and 15 MPa m/s at 2 m/s [20].

Friedrich et al. [21] pointed out the achievement of excellent tribological performance in PEEK-based composites loaded with both fibers and solid lubricant particles. A very low coefficient of friction and wear rate was obtained with a high value of the factor *P*⋅*V* = 28 MPa m/s. Qi et al. investigated friction and wear of polyimide tribocomposites when rubbing against various metal alloys, i.e., aluminum alloy, bronze and bearing steel. Results demonstrated that when sliding against bronze, the wear rate of polyimide reinforced with aramid particles and PTFE was almost one order of magnitude lower than those obtained at rubbing against aluminum alloy and bearing steel. The composite exhibited much faster wear when rubbing against aluminum alloy under elevated *P*⋅*V* conditions. Comprehensive analyses of the tribo-chemistry indicated that radicals deriving from broken PTFE molecules were more prone to chelate with the bronze, thus forming a robust tribofilm [22].

Numerous studies of the tribological characteristics of PI-based composites have shown that the necessary condition for improving wear resistance is the formation of a thin and uniform transfer film (TF) and its reliable adhering on wear track surfaces of both counterpart and polymer composite, regardless of the loaded filler types [23,24,25,26,27].

With respect to the sliding of polymer composites, tribofilm formation is a complex process governed by both tribophysical and chemical actions [28,29,30,31]. The reactions of PPS with nanofillers, e.g., nano-CuO and TiO_2_, and the steel counterface, were also observed by Bahadur and Sunkara [18]. With respect to the tribofilms of the nano-SiO_2_/SCF/EP composites, a glassy tribofilm consisting of silica, iron oxide and carbon (C)-material was identified [31,32]. Harris et al. reported that when PTFE/nano-Al_2_O_3_ composites was slid against 304 stainless steel, broken PTFE chains chelated to both the metal counterface and the alumina nanoparticles, which was stimulated the generation of a stable lubricating tribofilm [28].

It was inferred that tribosintering of wear products, i.e., released nanoparticles, tribo-oxidation product, and remnant material from the matrix, dominated the tribofilm formation [30,31,32]. Zhang et al. suggested that a possible easy-to-shear characteristic of the tribofilm could be responsible for the high tribological performance especially under high *P*⋅*V* conditions [31,32].

In such a context, the issue of the TF formation and fixation is decisive in ensuring the performance of antifriction composites based on high performance polymers (HPP), including both PI and PEI. Tribochemical reactions of polymer matrices with metallic counterparts were deemed important in determining tribofilm formation and the tribological performance [33,34,35,36,37]. Gong et al. [38] proposed that PTFE molecules chemically bonded onto metallic counterfaces, e.g., iron (Fe), aluminum (Al) and zinc (Zn). Gao [39] revealed that PTFE reacted with rubbing steel in air ambience owing to the presence of oxygen and humidity. Besides PTFE, tribochemistry of polyphenylene sulfide (PPS) also gained interest for fundamental research. Bahadur et al. [40] suggested that the PPS can react with steel counterface and metallic oxide fillers, which were expected to enhance the adhesion between tribofilm and a counterpart.

Huimin Qi [41] comparatively studied tribochemistry of ultrahigh molecular weight polyethylene (UHMWPE), polyphenylene sulfide (PPS) and polyetherimide (PEI) in tribocomposites. It was revealed that when rubbing with steel, molecular chains of UHMWPE were broken and free radicals finally chelated with the counterface. Whereas, PPS molecules underwent thermal decomposition, oxidation and finally ferrous sulfide (FeS) and ferric sulfate [Fe_2_(SO_4_)_3_] were generated. Tribochemistry of polymer molecules played an important role in tribofilm formation and tribological performance of the conventional composites. When sliding over the hybrid nanocomposites at low *P*⋅*V* conditions, tribochemistry of polymer molecules played a similar role as for the conventional composites. Nonetheless, at high *P*⋅*V* products, independent of the polymer matrices, robust tribofilms containing a high fraction of silica were adhered on the steel counterface, minimizing direct rubbing of the friction pair.

It was found in [42,43] that the molecular chain conformation strongly alters the transfer film properties: (i) highly crystalline contents [e.g., a-crystalline polyamide (PA)] are unfavorable for smooth transfer; whereas (ii) orientation into a rigid amorphous phase [e.g., poly(ethylene terephthalate) (PET)] and polycondensation reactions [e.g., polyimide (PI)] promote the formation of a smooth transfer film.

It was believed that tribochemical reactions of polymers were rather complex with multiple influential parameters. First, polymer chain structures play a role since scission of molecular chains can lead to the formation of free radicals [28,37]. From the aspect of tribochemistry, such active elements as sulfur and fluorine in polymer structure, possibly lead to the generation of sulfide and fluoride, respectively [28,40]. In addition, functional fillers can also take part in the tribochemistry with polymer matrices and exert an influence on the tribological performance. In addition, stress and temperature flashes on the friction interface are deemed to be essential for activating various tribochemical reactions. Although the importance of tribochemistry of polymers was recognized, comprehensive understanding on effects of molecular structures of polymer matrices and material formulations is still lacking.

In this paper, which continues the ideas stated in [44], the tribological characteristics of reinforced antifriction both PI- and PEI-based composites were estimated on different (metal and ceramic) counterfaces in a linear tribological contact. The key problems of interest were as follows: (i) What might improve the wear resistance of three-component composites since the presence of enforcing CF prevents transfer film formation? (ii) Are there any advantages of using less expensive PEI-based composites in terms of tribological behavior? (iii) How does the type of polymer matrix affect the Wear rate (WR) and Coefficient of Friction (CoF) within the wide span of *P*⋅*V* product?

## 2. Results

The mechanical properties of the neat polymers, as well as the PI- and PEI-based composites filled with identical reinforcing (CCF) and modifying polymer (PTFE [45]) and crystalline (Gr, MoS_2_ [46]) fillers differed slightly (Table 1). The difference in the mechanical characteristics of the composites based on both matrices was leveled by the CCF reinforcing effect. The elongation at break values of all composites decreased by a factor of 4–5, and their fracture type was brittle [44].

Before presenting the results of the tribological tests, the authors considered it appropriate to formulate three hypotheses, on the basis of which the interpretation of the obtained data was carried out.

1. Improving wear resistance of high-strength polymer materials is traditionally associated with the TF formation, which smooths out asperities and effectively reduces roughness on the harder counterpart surface. Additionally, a TF contains tribologically degraded and oxidized products, adhered on the more rigid counterpart surface, which are able to suppress the chemical interaction of the mating parts. Another transfer layer (tribofilm) is formed on the composite wear track surface, which also mainly consists of debris. To designate it, in addition to the generally accepted ‘transfer film layer’ term [47], the authors propose to use the ‘tribological layer’ (TL) one, which actually represents ‘mechanically mixed layer’ or mechanically mixing debris with contaminants [48]. It is assumed that both fatigue and ratcheting wear mechanisms are involved in the TL formation process. In this paper, the outer rings of a rolling bearing with the sufficiently high R_a_ roughness of 0.20–0.25 µm were used as a counterface. With its diameter of 35 mm, the ring circumference was ~110 mm. In this case, a new counterpart segment (a ‘renewed’ surface) comes into contact with the wear track at each moment in time, in contrast to the ‘pin-on-disk’ scheme (when the counterface contact area is unchanged and, as a rule, does not exceed 10 mm^2^). In combination with the scraping action of enforcing CF, it is very difficult to form and reliably adhere a TF on a counterpart of such a large area and a small wear track (as a solid lubrication and debris source). Consequently, improving wear resistance can predominantly be achieved by the TL formation on the composite wear track surface.

2. Evaluation of the influence of the TL formation process and its characteristics on wear resistance can be carried out both quantitatively and qualitatively. On the one hand, a smooth topography of a structurally homogeneous TF on a wear track surface qualitatively confirms the formation of a similar TL. However, a WR value is measured only at the end of a tribological test, as an integral assessment of the interaction of the mating materials. At the same time, the TL formation is a dynamic process that can possess a different duration. In addition, the TL can be carried away and re-formed. Thus, one assessment of the WR value correlates with the facts and features of the TL formation process, but does not enable us find out all its patterns and their effects.

3. High strength polymers (primarily HPP) possess great CoF levels. The TL formation processes may be accompanied by their noticeable lowering. Thereby, an analysis of the CoF time dependences under different conditions of tribological tests should enable us to trace in detail the TL formation kinetics. It is assumed that reaching a low and steady CoF value is a reliable indicator of the TL formation on a composite wear track surface. Confirmation of this fact should be sought in micrographs characterizing its topology.

### 2.1. The Ceramic-Polymer Tribological Contact

Since the ZrO_2_ counterpart was inert with respect to both polymers, initially the tribological studies were carried out for the ceramic-polymer tribological contact. Figure 1 shows the tribological characteristics of the PI- and PEI-based composites at various *P*⋅*V* load-speed modes (within the *P* load range of 60–180 N and the *V* sliding speed values of 0.1–0.5 m/s). For these test conditions, the P_max_ maximum contact pressure was estimated to be 66.9 MPa for *P* = 60 N; 86.3 MPa for *P* = 100 N; 102.1 MPa for *P* = 140 N; and 115.8 MPa for *P* = 180 N. It followed from Figure 1 that there was a fundamentally different pattern of the effect of the solid lubricant fillers on the tribological properties of the composites relative to those for the ‘ball-on-disk’ scheme [44], when loading with PTFE provided the maximum wear resistance.

Under the ‘mild’ tribological conditions (*P* = 60 N and *V* = 0.1 m/s), WR values were rather low and comparable for neat PI and PEI (~4 × 10^−6^ and ~2 × 10^−6^ mm^3^/N⋅m, respectively). They increased with the raising of the *P*⋅*V* product, but the process was much more intensive in the neat PI case. Under the ‘severe’ conditions (*P* = 180 N and *V* = 0.5 m/s), the WR value of neat PEI was 30 times less than that for neat PI. A CoF level of 0.5 was observed for neat PI throughout the studied *P*⋅*V* range. At the same time, it was lower for neat PEI and sharply decreased down to 0.18 (Figure 1c,d) only under the ‘severe’ conditions. In this case, both PI and PEI surface temperatures were quite high, but it was 1.9 times lower on the neat PEI sample (100 °C versus 190 °C, Figure 1e,f).

On the other hand, the PI/10 CCF composite possessed greater wear resistance over the entire studied *P*⋅*V* range, compared to that for the PEI/10 CCF one. Additional loading with the Gr and MoS_2_ fillers exerted an ambiguous effect on this parameter by changing the *P*⋅*V* level for the PI- and PEI-based composites. For the PI-based ones, Gr was more efficient at low *P*⋅*V* values (up to 40 N⋅m/s), while MoS_2_ showed a better effect above this level. Under these test conditions, the CoF level of the ternary composites based on both matrices was close (about 0.1 according to Figure 1c,d). A similar pattern was observed for their surface temperatures, although it was evidently lower for the PEI/CF sample (Figure 1e,f).

Note that PTFE was not effective filler for the composites based on both matrices in the ceramic-polymer tribological contact. According to the authors, this was due to the fact that it did not contribute to the formation of a stable TL on the composite wear track surfaces (discussed in more detail below).

In doing so, the chemical nature of PI and PEI played a decisive role in the tribological characteristics, mainly for the neat polymers in the ceramic-polymer tribological contact in the entire studied range of the *P*⋅*V* product. For the PI/10CCF and PEI/10CCF composites, additional loading with the Gr and MoS_2_ fillers determined the different pattern of wear resistance at the low *P*⋅*V* levels (up to 40 N⋅m/s) through the different pattern of the TL formation. Above this *P*⋅*V* level, wear resistances of the binary and ternary composites were close in magnitude at the nearly equivalent CoF values.

Below, the wear track surfaces and the CoF time dependences are presented for the studied materials tested under the ‘mild’ (Figure 2) and ‘severe’ (Figure 3) conditions. Their analysis from the standpoint of the TL formation enabled us to evaluate the key factors influencing their structures as a factor determining wear resistance. It should be noted that the solid lubricant fillers did not perform their function, but rather played the role of a TL structure modifier under the applied tribological test conditions (the linear contact, high roughness, low specific pressure and large counterpart sliding area). In particular, solid lubricant particles were absent in the PI/10CCF composite case. In addition, its CoF level was very low (~0.12 under the ‘mild’ conditions and ~0.1 under the ‘severe’ ones, Figure 2f and Figure 3f, respectively). In general, the mechanism of the TF formation on the counterpart, as a way of imparting antifriction properties, was not involved.

Under the ‘mild’ tribological conditions, fractured CCFs were found on the wear track surface of the PI/10CCF composite (Figure 2e). In this case, a CoF level quickly reduced down to ~0.12 (Figure 2f) with a rather negligible WR value of ~2.6 × 10^−6^ mm^3^/N⋅m (see Table 2 below). According to the above-mentioned hypotheses, this meant that a TL was formed, being reinforced with fractured CCFs (see Figure 2e). Such a CoF evolution over time showed that it did not have a long running-in period at the first stage (in a test distance up to 200 m), when CoF values typically increased. As a result, the CoF level was low and stable over time as it declined from the initially high value. On the contrary, a WR value of ~10.3 × 10^−6^ mm^3^/N⋅m was observed at a great CoF level of 0.37 (Figure 2h) for the PEI/10CCF composite, i.e., a TL was not formed on its wear track surface (Figure 2g).

The loading with PTFE of the PI/10CCF and PEI/10CCF composites was accompanied by a fold increase in WR values with high CoF oscillation levels (Appendix A). The topography of the wear track surfaces also did not indicate the TL formation fact (Appendix A). In a previous paper [49], the authors showed that loading PI with PTFE caused the formation of an inhomogeneous structure and the absence of adhesion between the polymer components.

On the contrary, loading with Gr and MoS_2_ as the third component changed the topography of the wear track surfaces. For both types of the polymer matrices, WR values decreased down to ~0.31 × 10^−6^ and ~1.85 × 10^−6^ mm^3^/N⋅m for the PI/10CCF/10Gr and PEI/10CCF/10Gr composites, respectively, at CoF levels around 0.2 (Figure 2j,l). Their wear track surfaces were smooth (Figure 2i,k). Judging by the CoF kinetics (a gradual decrease with reaching the stable low level), a solid TL was formed. This meant that the composition and structure of the composites provided the possibility of its formation and adhering. Recall that a topographically similar TL was also formed on the PI/10CCF composite (Figure 2e), but the WR level was 2.6 × 10^−6^ mm^3^/N⋅m (Table 2). The reason for this fact might be the high volumetric wear at the TL formation stage, i.e., at the test distance from 0 up to 200 m.

In other words, under the ‘mild’ tribological conditions in the linear contact, the TL was formed effectively in the case of the PI/10CCF composite with the less flexible molecular structure, but it was not found when the PEI/10CCF one was examined. The Gr addition in the PI/10CCF and PEI/10CCF composites provided the most efficient TL formation that enabled the lowering of both WR and CoF levels (Table 2).

Under the ‘severe’ conditions, the surface temperature enhanced sharply up to T = 180 °C in the neat PI case, due to a low heat dissipation of the ceramic counterpart. Melting of the material on the wear track surface was observed (which was accompanied by a high level and oscillating nature of the CoF changes around 0.51). For neat PEI, on the contrary, its CoF value decreased down to 0.17 (Figure 3a–d). To some extent, this trend of the CoF time dependence (Figure 3d) testified in favor of the TL formation, but it could not be stated that it had the form of a continuous homogeneous film (Figure 3c).

Reinforcement of both polymers with CCFs was accompanied by reducing temperature in tribological contacts that caused a decrease in both WR and CoF levels to an equal extent (2.86 × 10^−6^ and 5.5 × 10^−6^ mm^3^/N⋅m; 0.1 and 0.1; Figure 3f and h, respectively). On the wear track surfaces of both PI/10CCF and PEI/10CCF composites, fractured CCFs were evident, which manifested itself in high surface R_a_ roughness of the wear track surfaces (0.251 and 0.303 µm, respectively, see Table 2), and a decrease in CoF values (from 0.4 down to 0.1 in this case) indicated the TL formation, reinforced with CCFs (Figure 3e,g). At the same time, the TF was not a continuous homogeneous film due to the ‘severe’ tribological conditions. Note that the WR value was not fully an indicator of the TL formation, since intense wear at the initial stage of the tribological tests (when CoF values gradually decreased to a consistently low level at a distance of 0–200 m) was responsible for overall WR enhancing.

In regard to three-component composites, it was shown that loading with Gr was almost equally as efficient for both types of matrices. Both WR and CoF levels, as well as surface R_a_ roughness, were close in magnitude (Figure 3j,l). Note that the ‘severe’ tribological conditions prevented the formation of a continuous uniform TL (Figure 3i,k). Thus, it was the ability to form and adhere a TL that determined the antifriction properties of the two- and three-component composites based on both PI and PEI thermoplastic matrices in the ceramic-polymer tribological contact under the ‘severe’ conditions. In the cases of the PI/10CCF, PEI/10CCF, PI/10CCF/Gr and PEI/10CCF/Gr composites, the TL formation patterns were of similarity.

### 2.2. The Metal-Polymer Tribological Contact

Changing the counterpart material on the metal one could be accompanied by the following variations in the test conditions. Firstly, the steel counterface well dissipated heat from the tribological contact zone due to its high thermal conductivity. Secondly, the steel counterpart could be damaged by friction against CCFs. Thirdly, the ball bearing steel was more chemically reactive towards the polymers, especially under the ‘severe’ tribological conditions. This fact was reflected in dependences of the tribological characteristics from the *P*⋅*V* product (Figure 4). The following identified features should be noted. Firstly, there were high WR values for both neat PI and PEI even under the ‘mild’ tribological conditions, but it was several times higher for neat PI compared to that for neat PEI, as was also found in the ceramic counterpart case (Figure 4a,b). A CoF value was also greater for neat PI than that for neat PEI. It was about 0.5–0.6 for neat PI over the entire studied *P*⋅*V* range (Figure 4c), while it was lower and varied less than 0.4–0.45 for neat PEI (Figure 4d).

Secondly, loading with all studied types of the fillers (fibers and particles), excluding PTFE, equally reduced CoF values with rising ‘severity’ of the tribological conditions. Surface temperatures were lower on all investigated composites in the metal-polymer tribological contacts (Figure 4e,f) compared to those in the ceramic-polymer ones, which was determined by a higher thermal conductivity of the steel (see Table 3).

Thirdly, PTFE particles did not provide solid lubrication in the entire studied *P*⋅*V* range, and was much as it was observed for the ceramic-polymer tribological contact. Such composites showed WF values close to that of neat polymers. However, CoF levels of the three-component composites filled with PTFE were close to those for the CCF-reinforced ones loaded with the Gr and MoS_2_ fillers. A WR value of the PI/10CCF/10PTFE composite was almost the same as that of neat PI over the entire *P*⋅*V* range, while it was ~20 × 10^−6^ mm^3^/N⋅m for the PEI/10CCF/10PTFE one, and even exceeded that of neat PEI with raising the *P*⋅*V* parameter (Table 3).

Fourthly, additional loading of the PI/10CCF composite with the Gr and MoS_2_ fillers did not significantly increase WR values compared to that of the binary CCF-reinforced one under the ‘mild’ tribological conditions. However, they were efficient under the ‘severe’ ones. In this case, WR values were reduced by factors of three and sixty, respectively, compared to that of neat PI. Moreover, the influence of both crystalline fillers was identical (Figure 4a,b).

Fifthly, graphite was the most effective filler for the PEI-based composite at *P*⋅*V* < 40 N⋅m/s, while MoS_2_ was the most effective above this level. Since the CCF-reinforced composites both with and without solid lubricant particles possessed comparable tribological properties, it could be considered that the formed TL played a decisive role again.

Below is presented an analysis of the wear tack surface topography and the CoF time dependences under both ‘mild’ and ‘severe’ tribological conditions (Figure 5 and Figure 6, respectively). The goal was to identify the main factors that determined the TL structure.

In general, the pattern of the WF and CoF dependences on the *P*⋅*V* product in the metal-polymer tribological contact (Figure 5 and Figure 6) correlated with that for the ceramic-polymer one (Figure 2 and Figure 3). For neat PI and PEI, WR values (33.6 × 10^−6^ and 7.7 × 10^−6^ mm^3^/N m, respectively) and surface R_a_ roughness (0.389 and 0.098 µm) differed significantly at high CoF levels in the metal-polymer tribological contact under the ‘mild’ conditions (Figure 5a–d). Their surface temperatures were low due to the high thermal conductivity of the metal counterpart. As on the ceramic one, neat PEI showed a noticeably less WR value (Table 3).

Reinforcement with CCFs caused a fold decrease in both WR and CoF (0.26 versus 0.50) levels for the PI/10CCF composite, while a WR value of the PEI/10CCF one, on the contrary, increased with lowering its CoF level down to 0.35 with a pronounced oscillation (Figure 5e–h).

So, the effect of the polymer matrix type in the metal-polymer tribological contact under the ‘mild’ conditions was generally identical to that in the ceramic-polymer one. At the same time, the TL formation on the PI/10CF, PI/10CF/10Gr and PEI/10CF/10Gr composites occurred at the higher CoF level (0.25 instead of 0.20). Under the ‘severe’ tribological conditions (Figure 6d), a WR value for neat PEI was also half that for neat PI (32.6 × 10^−6^ and 68.6 × 10^−6^ mm^3^/N⋅m, respectively) at a lower CoF level (0.46 versus 0.52). The trend of its variation (Figure 6c) was similar to that in the ceramic-polymer tribological contact.

A clear trend towards the TL formation was observed for the CCF-reinforced composites. CoF values quickly reached stable low levels (0.17 versus 0.23, while it was 0.1 for both types of the polymer matrices in the ceramic-polymer tribological contacts). On the PEI/10CCF composite, its surface R_a_ roughness of 0.231 µm was also lower (Table 3).

Further filling with Gr and MoS_2_ also resulted in the TL formation. At the same time, CoF values of the three-component PEI-based composites were lower (0.16 and 0.15 versus 0.20 and 0.23 for the PEI- and PI-based ones, respectively, against the ~0.10 in the ceramic-polymer tribological contacts). On the other hand, TLs were more continuous in the metal-polymer tribological contacts (Figure 5i,k).

Thereby, the TL formation during friction of the PI/10CF, PI/10CF/10Gr and PI/10CF/10MoS_2_ composites on the steel counterpart occurred at higher CoF levels (0.20–0.23 instead of 0.10, see Appendix A). For the PEI/CF, PEI/CF/Gr and PEI/CF/MoS_2_ ones, the CoF values of 0.15–0.17 were lower (but were greater than upon friction against the ceramic counterpart). The latter result could be significantly influenced by the temperature in the tribological contact, which was noticeably higher for the ceramic counterpart.

The authors determined that roughness of both counterparts was about 0.150 µm and did not noticeably change for all the studied PI- and PEI-based composites under the applied tribological test conditions in the entire *P*⋅*V* range. This indicated the absence of the TF formation and adherence on the counterparts due to the relatively low contact pressures and the ‘non-renewable’ tribological contact surfaces of the polymer composites, as well as the counterface surface scraping by CCFs. Consequently, the only way to improve wear resistance was the TL formation.

### 2.3. Nanoindentation Tests

Below are the investigation results about formed TLs on the PI- and PEI-based composites loaded with reinforcing (CCF) and modifying crystalline (Gr) fillers obtained by the nanoindentation method. The only samples were examined after the tribological tests in the metal-polymer tribological contact under the ‘severe’ conditions. Figure 7 shows dependences of the *h* nanoindenter penetration depth, *H* hardness, the *E* elastic modulus, and the elastic recovery ratio for the wear tracks of both neat PI and PEI, as well as their composites from the load level. By increasing the load, the TL depth was indirectly estimated. Additionally, all values of these parameters are presented in Table 4, measured at the minimum, i.e., on the TL surface (provided that it was formed) and maximum loads. In four columns on the right the above data on their tribological testing are summarized as well.

It is seen that *H* hardness (1.02 GPa) and the *E* elastic modulus (10.9 GPa) on the wear track surface of neat PI (where, in fact, a TL was not formed) were almost twice as high as the corresponding characteristics for neat PEI (*H* = 0.56 GPa and *E* = 6.5 GPa, respectively). In the second case, a TL was not formed as well (judging by the CoF time dependence). With an increase in the load on the indenter, these characteristics were leveled. This indicated a small thickness of the tribologically modified surface layer on neat polymers (according to the authors, its thickness did not exceed 500 nm); secondly. It is suggested that a debris film was formed on both polymers’ wear track, the presence of which reduced their antifriction characteristics (variations in their strength properties were associated with a different response to the impact of the counterpart).

Their elastic recovery ratios in the surface layers (0.48 and 0.41) in comparison with those for the PI/10CCF and PEI/10CCF composites (0.25 and 0.27) also testified in favor of extremely thin thicknesses of the tribologically modified layers on the wear track surfaces of both neat PI and PEI.

In the PI/10CCF and PEI/10CCF composites (where TLs was definitely formed), the *H* hardness values on the surface layers were close (0.52 and 0.50 GPa, respectively), and the *E* elastic moduli were almost equal to those for the neat polymers. Their elastic recovery ratios were also close to the neat polymers. In other words, the presence of reinforcing CCFs in PI and PEI did not substantially change the TL mechanical properties such as *H* hardness and the *E* elastic modulus. This could be due to their low content.

A different pattern was observed on the three-component composites filled with crystalline graphite. Both *H* hardness and the *E* elastic modulus values in the surface layer increased by 1.5 times on the PI/10CCF/10Gr composite and 2.0 times on the PEI/10CCF/10Gr one. In addition, both of these indicators were greater not only in the surface layers, but also in the underlying ones (according to the design of the nanoindenter pyramid, its penetration depth was 1.4 μm at *P* = 20 mN). Note that the elastic recovery ratio on the TL surface of the PEI/10CCF/10Gr composite (0.37) was significantly higher than that on the PI/10CCF/10Gr one (0.25), which indicated the influence of the chemical nature of the polymers on the TL formation and its elastic properties.

Hence, the obtained data indicated that the presence of dispersed graphite particles caused the formation of a more pronounced TL on such composites. However, according to the measured values of *H* hardness and the *E* elastic modulus, the polymer matrix type did not significantly affect its strength properties. This may be related to a similar magnitude of wear resistance.

### 2.4. SEM-Micrographs and EDS Analysis

Then, an EDS analysis was performed on the wear track surfaces after the tribological tests on the metal counterpart under the ‘severe’ conditions (Figure 8a–c and Figure 9a–c, Appendix A). The wear track surface on neat PI (where the TL formation was not detected) was found to be fractured and contained traces of both Fe and S, transferred from the metal counterpart. The wear track surface on neat PEI (in the absence of a TL) was smoother with no Fe traces (Figure 8d–f, Appendix A).

The PEI/10CCF/10Gr composite showed a significant Fe content on its wear track surface, which corresponded to both higher *H* hardness and the *E* elastic modulus (see Figure 7). The Fe transfer from the metal counterface was possible due to the damaging effect of CCFs, as well as the adherence of tribologically oxidized debris on the more active PEI. In general, the presence of Fe was typical in TLs on both PI/CCF and PEI/CCF composites at sliding on the metal counterpart. Thus, the matrices nature affected the TL formation and its adherence on the wear track surfaces of the PI/10CCF/10Gr and PEI/10CCF/10Gr composites, determining their performance characteristics.

Figure 9 shows SEM micrographs of the wear track cross-sections on the samples after the tribological tests, as well as the EDS analysis data (Appendix A). Firstly, it was extremely difficult to detect any thin TLs at the applied magnifications. For this reason, the SEM micrographs rather illustrated the material structure formed in the bulk samples. Secondly, Fe was absent in the areas located below the TL (in the composites loaded with CCFs, as well as both CCFs and Gr). Thirdly, CCFs were quite differently oriented in both PI and PEI matrices, so their fracture occurred during the tests (Figure 9b,e). Fourthly, the rather heterogeneous structure of the heat conducting Gr-containing composites should be more prone to the TL formation (Figure 9c,f), especially compared to the neat polymers (Figure 9a,d).

## 3. Discussion

The TL concept was traditionally used to interpret results of tribological tests of metals. In particular, the following definition was given in [48]: ‘Metallic wear occurs by two mechanisms.
Material may extrude out from the sides of the contact giving rise to thin slivers which, subsequently, separate to produce wear debris.The fracture of surface layer causes a piece of material to leave the surface as wear debris (‘delamination’ wear or ratcheting failure). The latter takes place when the accumulated deformation exceeds a critical value. The following effects take place [50,51]:
increase of hardness of the near-surface is a result of mixing with hard particles, chemical changes, and work hardening;the stress–strain behavior of the near-surface material may be changed; in doing so, the wear rate would alter;mechanical or chemical mixing during the formation of the Tribological Layer may give rise to changing the strain to failure of the surface material;the Young’s modulus of the material and the friction coefficient may change as well.’


According to the definition presented above, the object of study in this research was more consistent with the TL concept than the TF one, which was widely used by many authors on polymer tribology.

Since the studied materials can be qualitatively divided into three classes regardless of the polymer matrix type (the neat polymers, as well as the two-component CCF-reinforced composites and the three-component ones loaded with graphite), below are data for comparing the polymer matrix types and the counterpart materials in the form of Table 5, Table 6 and Table 7, as well as graphs in Figure 10.

It could be concluded from Figure 10 that:

1. For the neat polymers, when a TL was not actually formed, WR values depended significantly on the polymer type, the *P*⋅*V* product, and the counterpart material (Figure 10a). At the same time, CoF levels were constantly high, but they were lower in all cases for neat PEI with more flexible molecular chains (Figure 10b). For the same reason, WR values were greater for neat PI over the entire *P*⋅*V* range (Figure 10a). The counterpart material did not exert a critical effect on the behavior of the CoF time dependence (Figure 10b).

2. Loading with CCFs was the reason that the TL formed on the two-component PI-based composites over the entire *P*⋅*V* range, but only at *P*⋅*V* ≥ 30 N⋅m/s on the PEI-based ones (Figure 10c). The best indication of the TL formation was the CoF value reaching a constant minimum level (the optimum was 0.1 according to Figure 10d). At the same time, CoF values were up to two times lower on the ceramic counterpart than those on the metal one (the optimum was 0.1 versus 0.2, respectively).

3. In the cases of the three-component composites, the role of both polymer matrices was leveled in the entire *P*⋅*V* range due to the TL formation, and WF values did not exceed 5 × 10^−6^ mm^3^/N⋅m with a single exception (Figure 10e). At the same time, CoF levels were below 0.25 (Figure 10g), but actually did not reach the value of 0.1, which was typical for the PI/10CCF and PEI/10CCF composites (Figure 10d). Probably, the joint filling of the ‘mixed’ TL with CCFs and Gr microparticles increased its hardness (according to the nanoindentation data) and prevented both types of the counterparts from easier sliding. However, the reinforcement factor appeared to be prevalent over the CoF levels, so the three-component composites were more wear resistant (Figure 10e,c).

It is known that changes in loads and sliding speeds exert different effects on the wear intensity and, accordingly, its nature. For this reason, the WR = *f*(*P*⋅*V*) and CoF = *f*(*P*⋅*V*) dependencies are not always smooth. However, an important criterion for imparting antifriction properties to composites is a CoF down slope to a consistently low level (as a rule, 0.10 for a ceramic counterpart, or 0.15–0.20 for a metal one). As a result, through the TL formation, it can be judged how much a material is able to resist the wear effect of a counterpart in a linear tribological contact. It should also be highlighted that the material loss at the running-in stage (in the studied cases, during the TL formation) is typically limited in practical applications. Therefore, it is necessary to take into account and minimize WF values when choosing a composite for specific operating conditions.

As noted above, most of the papers on designing antifriction polymer composites are currently devoted to loading with nanoparticles and studying their role [47,52]. The authors of this article have carried out preliminary studies in this direction and have also found that the identified trend in the CoF time dependence is a sign of TL formation. It should be noted finally that the term ‘tribological layer’ used in this paper was borrowed from typical publications on the metal wear. In this case, the TL structure and composition may differ significantly from the wearing material. In polymer tribology, the term ‘transfer film’ is used, the meaning of which was more consistent with the fact that it is adhered to the surface of the counterpart. Taking into account the studies carried out, the authors considered the correct application of such a term, since the processes of ratcheting, cyclical impact of counterpart surface asperities, reinforcement with fractured CCFs, mixing of the worn counterpart metal in a tribological layer (in the case of a steel ring), and a change in mechanical properties developed under such conditions. This resulted in both the lowest CoF level of 0.1 for the ceramic counterpart and the extremely low WF value. For any practical implementations of such composites in tribological pairs, it also provides a short running-in stage at a minimal wear rate.

## 4. Materials and Methods

### 4.1. Materials

Both PEI and PI powders (the ‘Solver PEI ROOH’ and ‘PI-1600′ grades, Shanghai, China, Figure 11) were used with an average particle size of 16 µm. In the manufacture of composites, they were filled with the ‘Fluralit’ fine PTFE powder with a mean diameter of less than 3 μm obtained by the ‘F-4′ fluoroplastic thermal decomposition (‘Fluralit synthesis’ LLC, Moscow, Russia [46]); molybdenum disulfide (Climax Molybdenum, Phoenix, AZ, USA, ∅ 1–7 μm), as well as colloidal graphite (∅ 1–4 μm) [47]. Chopped carbon fibers (CCFs; Tenax^®^-A, Teijina Carbon Europe Gmbh, Heinsberg, Germany) with lengths of 2 mm (an aspect ratio of about 100) were loaded as reinforcing particles.

### 4.2. Fabrication of the Composites

The polymer powders and the fillers were mixed by dispersing the suspension components in alcohol using a ‘PSB-Gals 1335-05’ ultrasonic cleaner (‘PSB-Gals’ Ultrasonic equipment center, Moscow, Russia). The processing duration was 3 min at a generator frequency of 22 kHz. After mixing, the suspension was dried in a ‘Memmert UF 55’ drying cabinet (Memmert GmbH + Co. KG, Schwabach, Germany) with forced ventilation at a temperature of 120 °C for 3 h. Both PEI and PI-based composites were fabricated by hot pressing at a pressure of 15 MPa and a temperature of 370 °C with a subsequent cooling rate of 2 °C/min. Similar to the earlier study [44], contents of CCF and all three types of the solid lubricant fillers were chosen to be 10 wt.% (Table 8).

### 4.3. Physical and Mechanical Properties

Tensile properties of ‘dog-bone’ shaped samples were measured using an ‘Instron 5582′ electromechanical testing machine (Instron, Norwood, MA, USA). The samples were 64 mm in length with the gauge section dimensions of 10 × 3 × 2.6 mm. The required surface quality was ensured by polishing with sandpapers of various grit sizes (up to 1000). The number of samples of each type was at least four. Strains were assessed with a contact extensometer.

Nanoindentation was carried out with a ‘NanoTest’ system (Micro Materials Ltd., Wrexham, UK) using Berkovich diamond tips in the load-controlled mode. The maximum load was 50 mN. Both loading and unloading durations were 20 s. Holding durations at the maximum loading and 90% unloading levels (for thermal drift correction) were 10 and 60 s, respectively. Hardness (H) and reduced Young’s modulus (E) of the composites were determined from the load versus displacement curves using the Oliver–Pharr method [53].

### 4.4. Tribological Characteristics

Dry sliding friction tests were carried out according to the ‘block-on-ring’ scheme using a ‘2070 SMT-1’ friction testing machine (Tochpribor Production Association, Ivanovo, Russia). Load (*P*) levels on the samples were in the range of 60–180 N, while sliding speed (*V*) values were from 0.1 up to 0.5 m/s. The maximum Hertzian contact pressure (Pmax) levels were 66.9, 88.3, 102.1, and 115.8 MPa, respectively. A metal counterpart was made of the outer ring of a commercial bear (GCr15 steel) and a ceramic one was from ZrO_2_. They had a disk shape with a diameter of 35 mm and a width of 11 mm. Roughness (Ra) levels on the counterpart surfaces were 0.20–0.25 µm, which were evaluated with a ‘New View 6200’ profilometer (Zygo, Middlefield, CT, USA). The counterpart temperature was assessed using a ‘CEM DT-820’ non-contact infrared (IR) thermometer (Shenzhen Everbest Machinery Industry Co., Ltd., Shenzhen, China). WR levels were determined by measuring both width and depth of wear track surfaces according to stylus profilometry, followed by multiplication by its length. Then, they were calculated taking into account the load and sliding distance values:(1)$$\mathrm{Wear}\;\mathrm{rate}\;=\frac{\mathrm{volume}\;\mathrm{loss}\;\left({\mathrm{mm}}^{3}\right)}{\mathrm{load}\;\left(N\right)\times \mathrm{sliding}\;\mathrm{distance}\;\left(m\right)}$$

### 4.5. Structural Studies

The topography of the wear track surfaces was studied using a ‘Neophot 2’ optical microscope (Carl Zeiss, Jena, Germany) equipped with a ‘Canon EOS 550D’ digital camera (Canon Inc., Tokyo, Japan), and an ‘Alpha-Step IQ’ contact profiler (KLA-Tencor, Milpitas, CA, USA). The equation is enumerated.

The structural studies were carried out on cleaved surfaces of mechanically fractured notched specimens. Liquid nitrogen was used for cooling them down. A ‘LEO EVO 50’ scanning electron microscope (Carl Zeiss, Oberkochen, Germany) with an ‘Oxford INCA X-Max80’ attachment for EDS analysis (Oxford Instruments, Abingdon, Oxfordshire, England) was employed at an accelerating voltage of 20 kV.

## 5. Conclusions

Based on the obtained results, the following conclusions were drawn.

1. It was shown that the main difference in the wear pattern and intensity for the two types of the studied polymers (PI and PEI) was manifested for the neat materials. In addition to the polymer type, the wear intensity, whose range exceeded two orders of magnitude, was significantly affected by both the counterpart material and the load-speed parameters. However, the CoF levels were not lower than 0.4 (with the exception of neat PEI under the ‘severe’ conditions, when the CoF value was 0.2). On the surfaces of both neat PI and PEI, any TLs were not actually formed. With the close key strength properties, neat PEI, having a more flexible polymer chain, possessed the greatest wear resistance.

2. The two-component CCF-reinforced PI-based composites possessed the high anti-friction properties throughout the studied *P*⋅*V* range. For the PEI-based ones, the fold WR reduction was achieved at *P*⋅*V* ≥ 30 N⋅m/s, when the stable TL was formed. The key reason for the improved anti-friction properties of the two-component composites was the TL formation, which included protective reinforcing of broken CCFs. Moreover, CCFs in the composites partially damaged the steel counterpart, as a result of which Fe was found in the TL compositions, providing their additional ‘alloying’. For the two-component composites, the TL mechanical properties were close to those of the base materials. The TL formation in the CCF-reinforced composites eliminated the polymer matrix role, so their mechanical and tribological characteristics were comparable.

3. Since the solid lubricant free PI/10CCF and PEI/10CCF composites showed similar CoF levels compared to those for the three-component ones, it is claimed that solid lubricating particles did not perform their function. On the other hand, loading with PTFE particles was not accompanied by the TL formation, and was also due to the lack of bonding between PTFE and high-strength both PI and PEI. Even the presence of reinforcing CCFs did not protect such composites from intense wear, which was associated with the impossibility of the TF formation on the counterpart surfaces.

4. On the three-component composites loaded with Gr and MoS_2_, the TL formation was observed for both polymer types on both counterparts at all *P*⋅*V* values. After the Gr addition, the TL was thicker and had the higher strength properties due to its simultaneous effective reinforcement with broken CCFs and fine Gr particles (~4 µm in size). In this case, when filling with Gr, the minimum WF value was provided, typically less than (or equal to) those for the PI/10CCF and PEI/10CCF composites. Under metal-polymer tribological rubbing the Gr presence slightly lowered the CoF levels compared to those for the PI/10CCF and PEI/10CCF ones. However, this difference was leveled at the maximum *P*⋅*V* value, which was due to the TL formation as well.

Concerning practical applications, the PEI/10CCF/10Gr is recommended for metal-polymer tribological pairs (operating under dry sliding friction in linear contact in the studied range of loads and speeds *P* = 60–180 N, *V* = 0.1–0.5 m/s). In the case of ceramic-polymer tribological coupling, the PI/10Gr composite possesses the highest tribological performance. However, the PEI-based composites are preferable for industrial use due to better processability and low manufacturing costs.

## Figures and Tables

**Figure 1 molecules-27-06376-f001:**
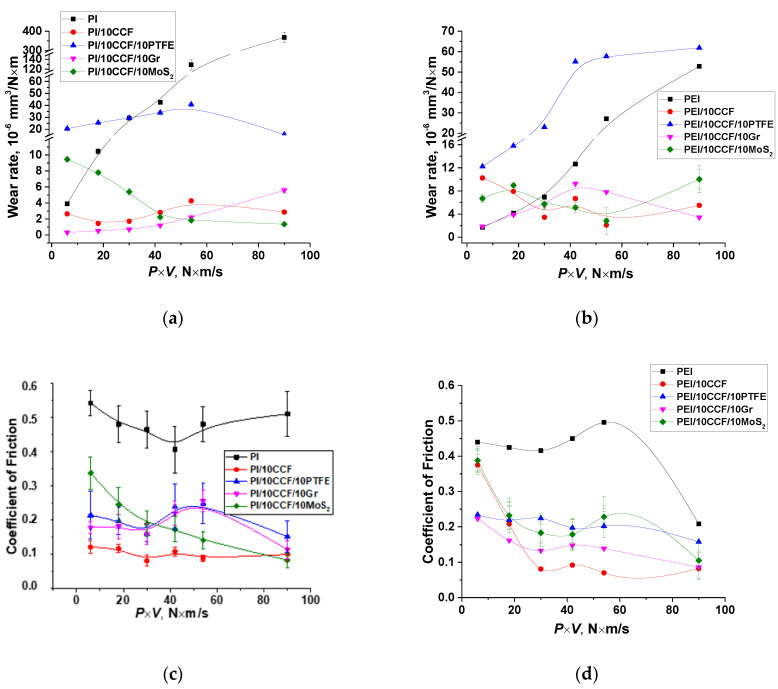
Dependences of the WR values (**a**,**b**), the CoF levels (**c**,**d**) and the surface temperatures (**e**,**f**) from the *P*⋅*V* load-speed parameter for the PI- (**a**,**c**,**e**) and PEI-based (**b**,**d**,**f**) materials. The ceramic-polymer tribological contact (the ‘block-on-ring’ scheme).

**Figure 2 molecules-27-06376-f002:**
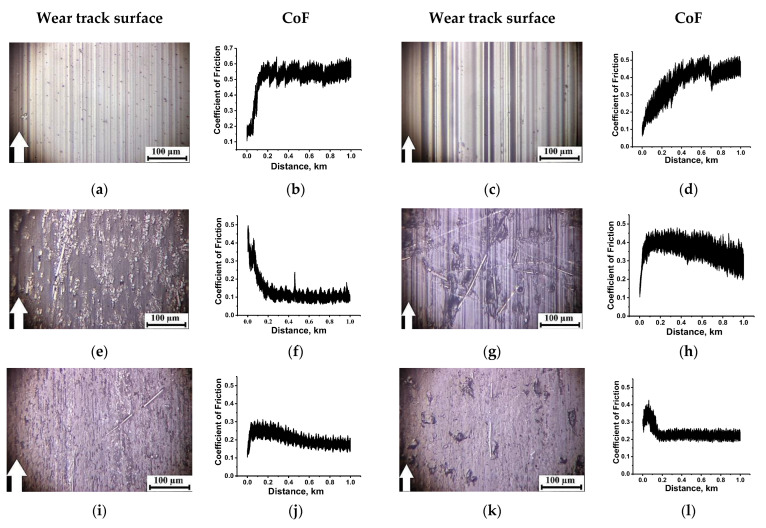
The optical micrographs of the wear track surfaces and the CoF time dependences for neat both PI (**a**,**b**) and PEI (**c**,**d**), as well as their PI/10CCF (**e**,**f**), PI/10CCF/10Gr (**i**,**j**), PEI/10CCF (**g**,**h**), PEI/10CCF/10Gr (**k**,**l**) composites. The ceramic-polymer tribological contact under the ‘mild’ conditions (*P* = 60 N, *V* = 0.1 m/s). See Appendix A for more details.

**Figure 3 molecules-27-06376-f003:**
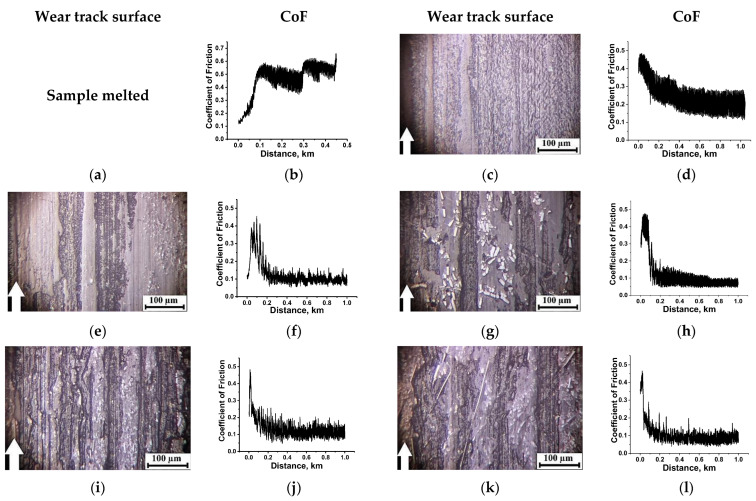
The optical micrographs of the wear track surfaces and the CoF time dependences for both neat PI (**a**,**b**) and PEI (**c**,**d**), as well as their PI/10CCF (**e**,**f**), PI/10CCF/10Gr (**i**,**j**), PEI/10CCF (**g**,**h**), PEI/10CCF/10Gr (**k**,**l**) composites. The ceramic-polymer tribological contact under the ‘severe’ conditions (*P* = 180 N, *V* = 0.5 m/s). See Appendix A for more details.

**Figure 4 molecules-27-06376-f004:**
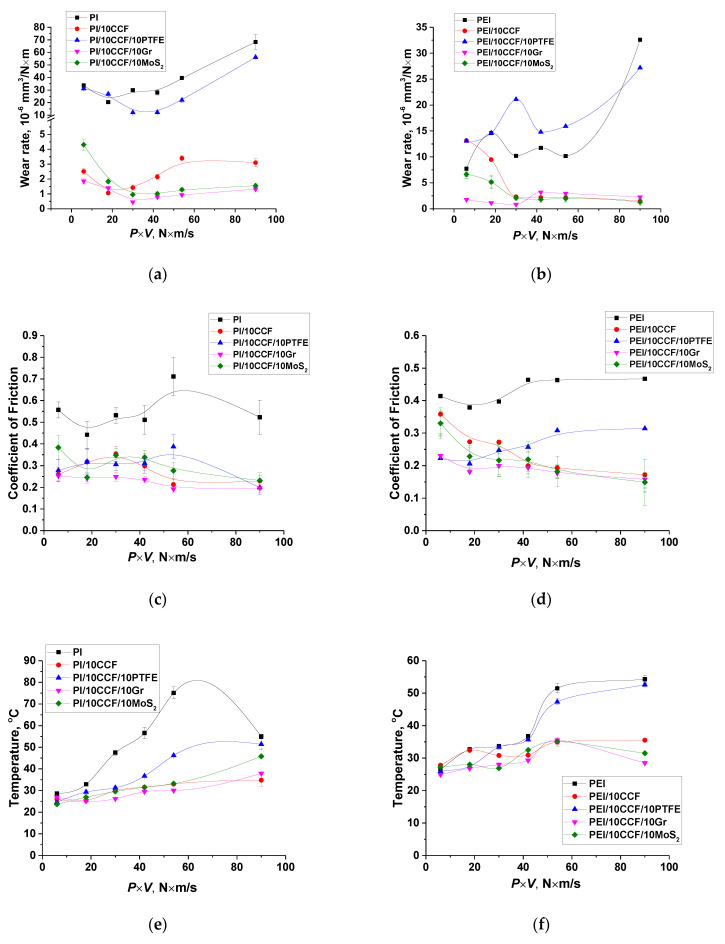
Dependences of the WR values (**a**,**b**), the CoF levels (**c**,**d**) and the surface temperatures (**e**,**f**) from the *P*⋅*V* load-speed parameter for the PI- (**a**,**c**,**e**) and PEI-based (**b**,**d**,**f**) materials. The metal-polymer tribological contact (the ‘block-on-ring’ scheme).

**Figure 5 molecules-27-06376-f005:**
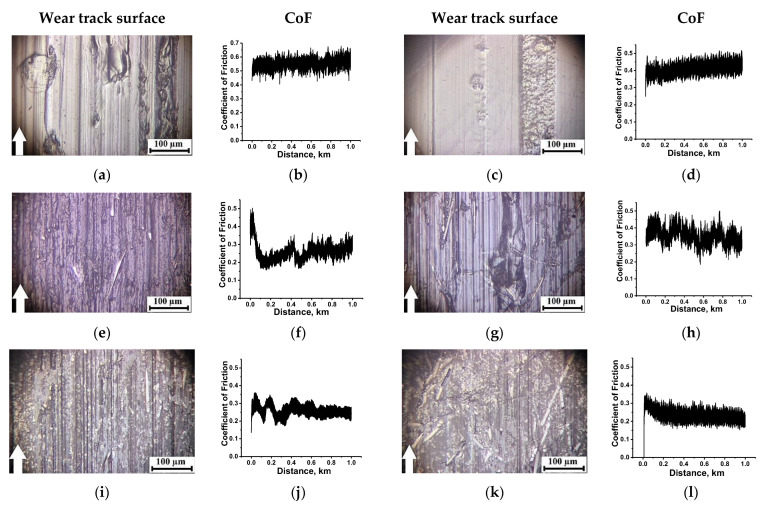
The optical micrographs of the wear track surfaces and the CoF time dependences for both neat PI (**a**,**b**) and PEI (**c**,**d**), as well as their PI/10CCF (**e**,**f**), PI/10CCF/10Gr (**i**,**j**), PEI/10CCF (**g**,**h**), PEI/10CCF/10Gr (**k**,**l**) composites. The metal-polymer tribological contact under the ‘mild’ conditions (*P* = 60 N, *V* = 0.1 m/s). See Appendix A for more details.

**Figure 6 molecules-27-06376-f006:**
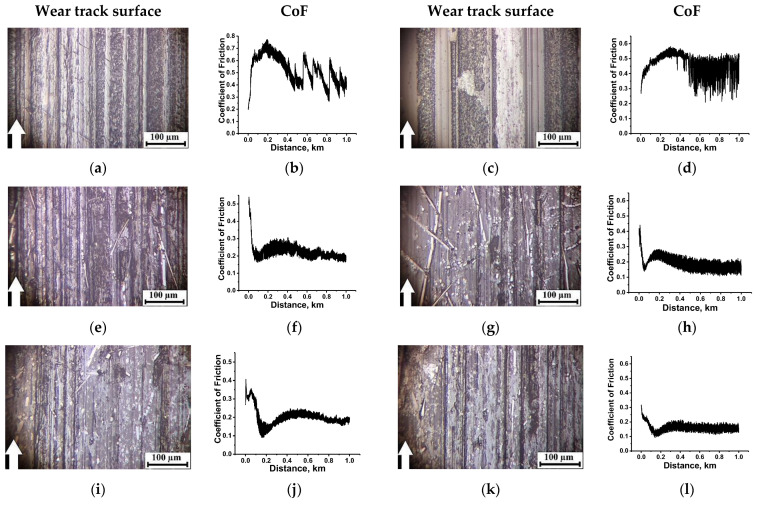
The optical micrographs of the wear track surfaces and the CoF time dependences for both neat PI (**a**,**b**) and PEI (**c**,**d**), as well as their PI/10CCF (**e**,**f**), PI/10CCF/10Gr (**i**,**j**), PEI/10CCF (**g**,**h**), PEI/10CCF/10Gr (**k**,**l**) composites. The metal-polymer tribological contact under the ‘severe’ conditions (*P* = 180 N, *V* = 0.5 m/s). See Appendix A for more details.

**Figure 7 molecules-27-06376-f007:**
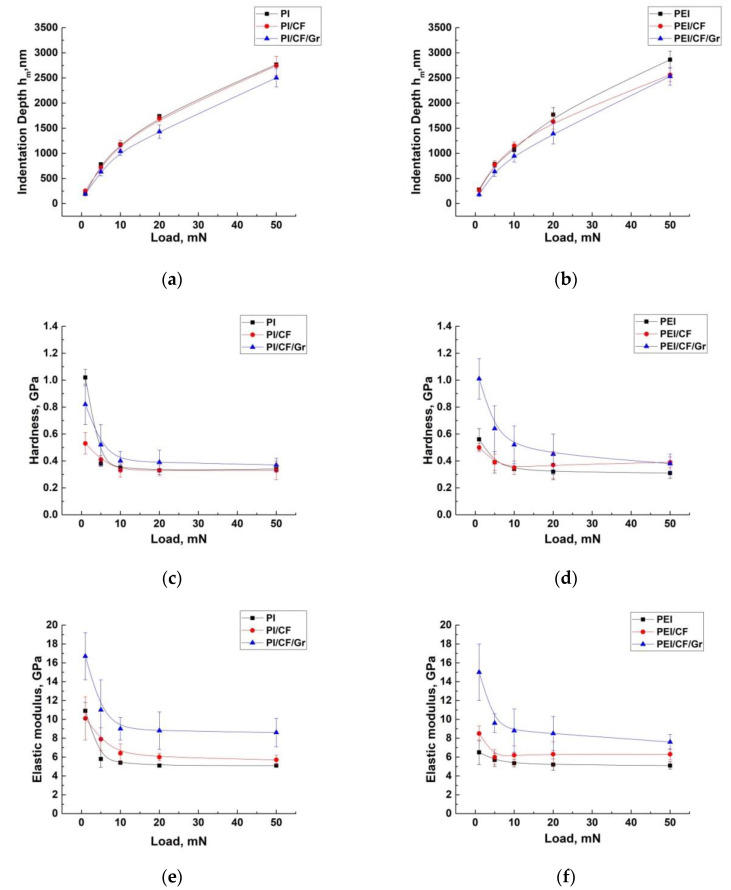
The dependences of the *h* nanoindenter penetration depth (**a**,**b**), *H* hardness (**c**,**d**), the *E* elastic modulus (**e**,**f**), and the elastic recovery ratio (**g**,**h**) for the wear tracks on both neat PI and PEI, as well as their composites from the load level.

**Figure 8 molecules-27-06376-f008:**
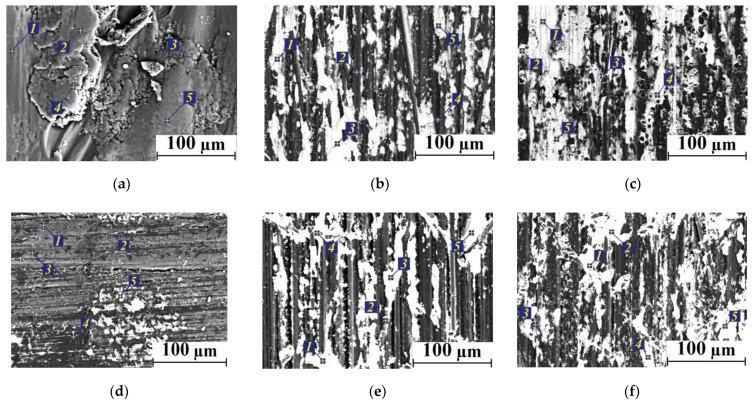
The SEM micrographs of the wear track surfaces after the tribological tests of neat PI (**a**), PEI (**d**), as well as the PI/10CCF (**b**), PEI/10CCF (**e**) and PI/10CCF/10Gr (**c**), PE/10CCF/10Gr (**f**) composites.

**Figure 9 molecules-27-06376-f009:**
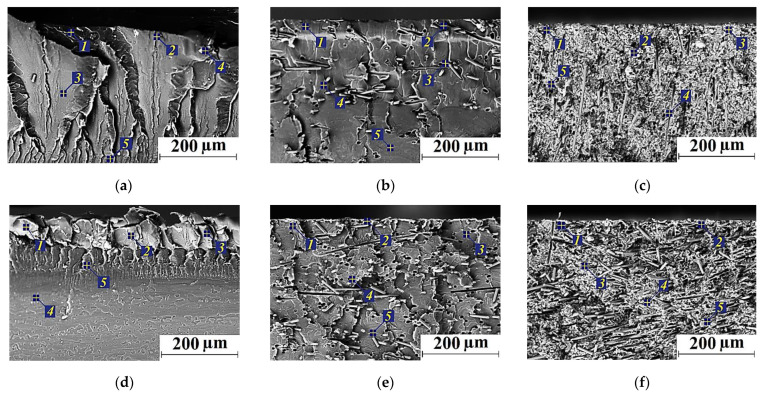
The SEM micrographs of the wear track cross-sections after the tribological tests of neat PI (**a**), PEI (**d**), as well as the PI/10CCF (**b**), PEI/10CCF (**e**) and PI/10CCF/10Gr (**c**), PEI/10CCF/10Gr (**f**) composites.

**Figure 10 molecules-27-06376-f010:**
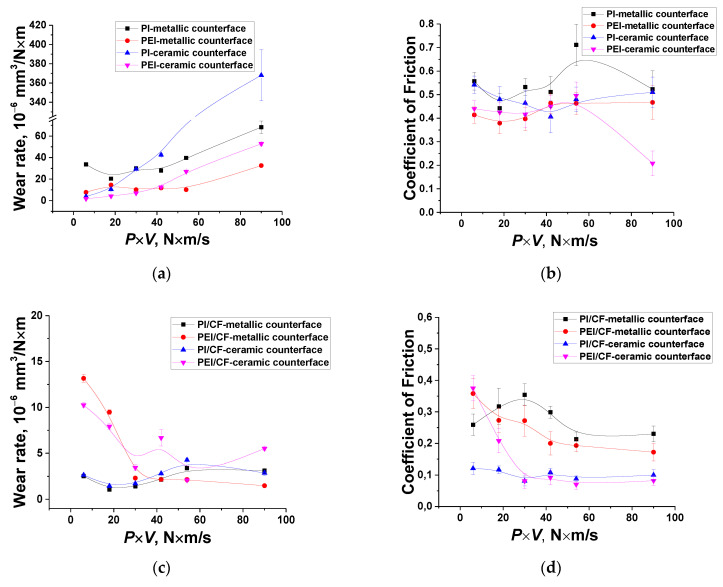
The dependence of the WR (**a**,**c**,**e**) and CoF (**b**,**d**,**f**) values from the *P*·*V* product for the PI- and PEI-based composites (the ‘block-on-ring’ scheme).

**Figure 11 molecules-27-06376-f011:**
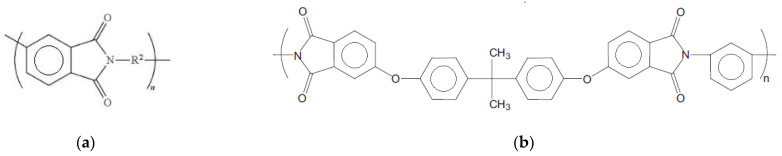
The PI (**a**) and PEI (**b**) molecular structures.

**Table 1 molecules-27-06376-t001:** The physical and mechanical properties of neat PI and PEI, as well as their composites.

No.	Filler Composition (wt.%)	Density *ρ*, (g/cm^3^)	Shore *D* Hardness	Elastic Modulus E (GPa)	Ultimate Tensile Strength σ_U_ (MPa)	Elongation at Break ε (%)
1	PI	1.37	80.2 ± 0.8	2.60 ± 0.69	110.7 ± 1	13 ± 0.7
2	PI/10CCF	1.42	80.6 ± 0.4	6.40 ± 0.33	152.1 ± 6.4	5.9 ± 0.3
3	PI/10CCF/10PTFE	1.44	77.5 ± 0.6	5.79 ± 0.45	115.9 ± 10.8	4.1 ± 0.3
4	PI/10CCF/10Gr	1.46	80.1 ± 0.3	6.35 ± 0.24	105.0 ± 3.9	2.7 ± 0.1
5	PI/10CCF/10MoS_2_	1.51	82.0 ± 0.3	6.06 ± 0.32	113.1 ± 9.1	3.0 ± 0.1
6	PEI	1.26	79.9 ± 0.3	3.12 ± 0.15	123.1 ± 0.5	16.1 ± 1.2
7	PEI/10CCF	1.31	81.4 ± 0.3	6.54 ± 0.43	153.2 ± 12.5	3.7 ± 0.6
8	PEI/10CCF/10PTFE	1.36	79.0 ± 0.3	6.17 ± 0.26	117.3 ± 8.0	3.1 ± 0.3
9	PEI/10CCF/10Gr	1.36	80.6 ± 0.2	6.37 ± 0.16	101.4 ± 2.6	2.8 ± 0.1
10	PEI/10CCF/10MoS_2_	1.41	81.9 ± 0.1	6.26 ± 0.17	121.0 ± 5.0	3.5 ± 0.3

**Table 2 molecules-27-06376-t002:** The ceramic-polymer tribological contact under the ‘mild’/‘severe’ conditions (“*P* = 60 N, *V* = 0.1 m/s”/“*P* = 180 N, *V* = 0.5 m/s”).

No.	Filler Composition (wt.%)	Average Wear Rate, mm^3^/N⋅m	Average Coefficient of Friction, ƒ	Average R_a_, µm	Average T, °C
1	PI	3.89 × 10^−6^/368.0 × 10^−6^	0.54/0.51	0.380/-	41.0/184.0
2	PI/10CCF	2.60 × 10^−6^/2.86 × 10^−6^	0.12/0.10	0.158/0.251	32.0/63.0
3	PI/10CCF/10PTFE	20.5 × 10^−6^/15.6 × 10^−6^	0.21/0.26	0.255/0.324	32.0/80.0
4	PI/10CCF/10Gr	0.31 × 10^−6^/5.6 × 10^−6^	0.18/0.10	0.112/0.436	31.9/70.0
5	PI/10CCF/10MoS_2_	9.45 × 10^−6^/1.34 × 10^−6^	0.34/0.08	0.286/0.256	35.3/52.8
6	PEI	1.72 × 10^−6^/52.8 × 10^−6^	0.44/0.17	0.431/0.201	38.0/100.0
7	PEI/10CCF	10.25 × 10^−6^/5.5 × 10^−6^	0.37/0.10	0.325/0.303	33.7/55.7
8	PEI/10CCF/10PTFE	12.2 × 10^−6^/61.8 × 10^−6^	0.23/0.15	0.395 /0.334	32.7/89.0
9	PEI/10CCF/10Gr	1.85 × 10^−6^/3.43 × 10^−6^	0.22/0.10	0.183/0.487	32.9/62.0
10	PEI/10CCF/10MoS_2_	6.7 × 10^−6^/10.0 × 10^−6^	0.38/0.10	0.418/0.509	35.6 /74.6

**Table 3 molecules-27-06376-t003:** The metal-polymer tribological contact under the ‘mild’/‘severe’ conditions (“*P* = 60 N, *V* = 0.1 m/s”/“*P* = 180 N, *V* = 0.5 m/s”).

No.	Filler Composition (wt.%)	Average WR, mm^3^/N⋅m	Average ƒ	Average R_a_, μm	Average T, °C
1	PI	33.6 × 10^−6^/68.3 × 10^−6^	0.50/0.52	0.389/0.250	28.6/55.0
2	PI/10CCF	2.5 × 10^−6^/3.1 × 10^−6^	0.26/0.23	0.505/0.333	26.0/34.0
3	PI/10CCF/10PTFE	31.5 × 10^−6^/56.1 × 10^−6^	0.28/0.20	0.446/0.297	24.6/51.0
4	PI/10CCF/10Gr	1.86 × 10^−6^/1.3 × 10^−6^	0.25/0.20	0.270/0.253	0.25/38.0
5	PI/10CCF/10MoS_2_	4.32 × 10^−6^/1.6 × 10^−6^	0.38/0.23	0.556/0.258	23.7/45.0
6	PEI	7.71 × 10^−6^/32.6 × 10^−6^	0.41/0.47	0.098/0.215	26.6 /54.0
7	PEI/10CCF	13.6 × 10^−6^/1.5 × 10^−6^	0.35/0.17	0.446/0.231	27.8/36.0
8	PEI/10CCF/10PTFE	13.4 × 10^−6^/27.2 × 10^−6^	0.22/0.31	0.352/0.188	25.9/53.0
9	PEI/10CCF/10Gr	1.7 × 10^−6^/2.3 × 10^−6^	0.23/0.16	0.260/0.229	24.9/29.0
10	PEI/10CCF/10MoS_2_	6.64 × 10^−6^/1.3 × 10^−6^	0.33/0.15	0.382/0.160	27.2/32.0

**Table 4 molecules-27-06376-t004:** The nanoindentation data for both neat PI and PEI, as well as their composites at the minimal/maximal load on the Berkovich pyramid.

Material	Hardness, GPa	Elastic Modulus, GPa	Elastic Recovery Ratio	WR, mm^3^/N⋅m	CoF	Temperature, °C	Roughness Ra, µm
Neat PI (no TL)	~1.02/~0.33	~10.9/~5.0	~0.48/~0.24	68.3 × 10^−^^6^	0.52	55	0.25
Neat PEI (no TL)	~0.56/~0.31	~6.5/~5.0	~0.41/~0.22	32.6 × 10^−^^6^	0.46	54	0.22
PI/10CCF	~0.52/~0.33	~10.0/~5.8	~0.25/~0.21	3.1 × 10^−^^6^	0.23	34	0.33
PEI/10CCF	~0.50/~0.39	~8.5/~6.3	~0.27/~0.22	1.5 × 10^−^^6^	0.17	36	0.23
PI/10CCF/10Gr	~0.81/~0.38	~16.6/~8.6	~0.25/~0.15	1.3 × 10^−^^6^	0.20	38	0.25
PEI/10CCF/10Gr	~1.02/~0.40	~15.0/~7.6	~0.36/~0.18	2.3 × 10^−^^6^	0.16	29	0.23

**Table 5 molecules-27-06376-t005:** A comparison of the tribological properties and the TL characters for neat PI and PEI.

Materials	WR, mm^3^/N⋅m	CoF	Temperature, °C	Roughness Ra, μm	TL Pattern	CoF Oscillation	CoF at the End of the Tests
**‘Mild’ tribological conditions (*P* = 60 N, *V* = 0.1 m/s)**							
Neat PI (ceramic)	3.89 × 10^−6^	0.54	41	0.38	no	Δ~0.15 weak breakdown	0.55
Neat PEI (ceramic)	1.72 × 10^−6^	0.44	38	0.43	no	Δ~0.15	0.45
Neat PI (metal)	33.6 × 10^−6^	0.50	28.6	0.39	no	Δ~0.15	0.6
Neat PEI (metal)	7.7 × 10^−6^	0.41	26.6	0.10	no	Δ~0.15	0.45
Comments: (1) a TL was not formed under the ‘mild’ tribological conditions; (2) WR values were several times greater on the metal counterpart at close CoF levels and lower temperatures. According to the authors, this was due to the activity of the counterpart material with respect to the polymer.							
**‘Severe’ tribological conditions (*P* = 180 N, *V* = 0.5 m/s)**							
Neat PI (ceramic)	368.0 × 10^−6^	0.51	184	melting	no	Δ~0.15 breakdown	0.50–0.60
Neat PEI (ceramic)	52.8 × 10^−6^	0.17	100	0.20	no	Δ~0.15	0.18
Neat PI (metal)	68.3 × 10^−6^	0.52	55	0.25	no	Δ0.60–0.30 low-frequency breakdown	0.40–0.50
Neat PEI (metal)	32.6 × 10^−6^	0.46	54	0.22	no	Δ~0.20	0.50
Comments: (1) a TL was not formed under the ‘severe’ tribological conditions as well; (2) on different counterparts, a variation in the WF values changed only by a factor of two for neat PEI, despite the high temperature (100 °C) in the ceramic-polymer tribological contact.							

**Table 6 molecules-27-06376-t006:** A comparison of the tribological properties and the TL characters for the PI/10CCF and PEI/10CCF composites.

Materials	WR, mm^3^/N⋅m	CoF	Temperature, °C	Roughness Ra, μm	TL Pattern	CoF Oscillation	CoF at the End of the Tests
**‘Mild’ tribological conditions (*P* = 60 N, *V* = 0.1 m/s)**							
PI/10CCF (ceramic)	2.6 × 10^−6^	0.22	32	0.16	yes	Δ < 0.10	0.10
PEI/10CCF (ceramic)	10.3 × 10^−6^	0.37	34	0.33	no	Δ > 0.15	0.28
PI/10CCF (metal)	2.5 × 10^−6^	0.26	26	0.51	yes	Δ~0.10	0.27
PEI/10CCF (metal)	13.6 × 10^−6^	0.35	27.8	0.45	no	Δ~0.15	0.30–0.40
Comments: (1) the stable TL formed on neat PI in the tribological test on the ceramic counterpart under the ‘mild’ conditions; (2) on the metal counterpart, the TL also formed on neat PEI and the identical WR value was reached at the higher CoF level; (3) a TL was not formed on the PEI-based composites. According to the authors, the TL reduced the interaction activity with the counterpart material.							
**‘Severe’ tribological conditions (*P* = 180 N, *V* = 0.5 m/s)**							
PI/10CCF (ceramic)	2.9 × 10^−6^	0.1	63	0.25	yes, fragmented	Δ < 0.10 breakdown	0.10
PEI/10CCF (ceramic)	5.5 × 10^−6^	0.1	56	0.30	yes, fragmented	Δ < 0.10	0.08
PI (metal)	3.1 × 10^−6^	0.23	34	0.33	yes, smooth	Δ < 0.05	0.20
PEI/10CCF (metal)	1.5 × 10^−6^	0.17	36	0.23	yes, smooth	Δ > 0.10	0.15
Comments: (1) under the ‘severe’ conditions, the TL formed on both PI/10CCF and PEI/10CCF composites; (2) in the test on the ceramic counterpart, the temperature was higher by ~20–30 °C, while the CoF level was lower (0.1). According to the authors, if TL had formed, WR values were comparable in magnitude for the PI/10CCF and PEI/10CCF composites under both ‘mild’ and ‘severe’ conditions.							

**Table 7 molecules-27-06376-t007:** A comparison of the tribological properties and the TL characters for the PI/10CCF/10Gr and PEI/10CCF/10Gr composites.

Materials	WR, mm^3^/N⋅m	CoF	Temperature, °C	Roughness Ra, μm	TL Pattern	CoF Oscillation	CoF at the End of the Tests
**‘Mild’ tribological conditions (*P* = 60 N, *V* = 0.1 m/s)**							
PI/10CCF/10Gr (ceramic)	0.3 × 10^−6^	0.18	32	0.11	yes, smooth	Δ~0.10	0.18
PEI/10CCF/10Gr (ceramic)	1.9 × 10^−6^	0.22	33	0.19	yes, smooth	Δ > 0.05	0.22
PI/10CCF/10Gr (metal)	1.9 × 10^−6^	0.25	27	0.27	yes, smooth	Δ~0.15 long running-in	0.24
PEI/10CCF/10Gr (metal)	1.8 × 10^−6^	0.23	25	0.26	yes, smooth	Δ~0.10	0.22
Comments: (1) under the ‘mild’ conditions, the patterns of changes in the WR and CoF values were similar for both types of the composites; (2) this phenomenon was different from that for the PI/10CCF and PEI/10CCF ones, as a stable TL was not formed on the PEI/10CCF wear track surface; (3) on the metal counterpart, the CoF levels were greater than those on the ceramic one.							
**‘Severe’ tribological conditions (*P* = 180 N, *V* = 0.5 m/s)**							
PI/10CCF/10Gr (ceramic)	5.6 × 10^−6^	0.10	70	0.44	yes, fragmented	Δ < 0.10	0.11
PEI/10CCF/10Gr (ceramic)	3.4 × 10^−6^	0.10	62	0.49	yes, fragmented	Δ < 0.10	0.10
PI/10CCF/10Gr (metal)	1.3 × 10^−6^	0.20	38	0.25	yes, smooth	Δ < 0.05	0.18
PEI/10CCF/10Gr (metal)	2.3 × 10^−6^	0.16	29	0.23	yes, smooth	Δ > 0.05	0.16
Comments: (1) under the ‘severe’ conditions, the TL formed on both types of the composites; (2) in the tribological test on the ceramic counterpart, the temperature was higher by ~20–30 °C, while the CoF level was lower (0.1) as in the cases of the two-component composites. According to the authors, the TL formation protected against wear under both ‘mild’ and ‘severe’ conditions.							

**Table 8 molecules-27-06376-t008:** Compositions and designations of the investigated polymer composites.

Filler Content, wt.%	Designation
PI	PI
PI + 10%CF	PI/10CCF
PI + 10%CF + 10%PTFE	PI/10CCF/10PTFE
PI + 10%CF + 10%Gr	PI/10CCF/10Gr
PI + 10%CF + 10%MoS_2_	PI/10CCF/10MoS_2_
PEI	PEI
PEI + 10%CF	PEI/10CCF
PEI + 10%CF + 10%PTFE	PEI/10CCF/10PTFE
PEI + 10%CF + 10%Gr	PEI/10CCF/10Gr
PEI + 10%CF + 10%MoS_2_	PEI/10CCF/10MoS_2_

## Data Availability

Not applicable.

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
