# Peer review of "Role of Testing Conditions in Formation of Tribological Layers at Line Contacts of Antifriction CF-Reinforced PI- and PEI-Based Composites†"

_molecules, 2022, doi:10.3390/molecules27196376_

Round 1

Reviewer 1 Report

In this paper, the authors compared and analyzed the two types of the high-strength PI and PEI- based polymers were carried out from the point of view of the tribological layer (TL) formation in the tribological tests according to the ‘block-on-ring’ scheme (a linear contact). This paper has certain innovations. In order to make it more logical and theoretical, this paper proposes the following suggestions

1. In the introduction part of this paper, it is not read coherently enough due to the lack of good logic. There should be some hierarchy and order in the questions about the background, need and current situation of the study. It is recommended to revise the logical order of the introduction.

2. According to the title of this paper, it is also important to analyze the mechanism of the effect of polymer matrices in the formation of tribological layers in line contacts.

3.  In the abstract, the authors claim: "The tribological layer formation patterns were determined by the chemical structure and flexibility of the polymer chains, the activity of the counterpart materials, the modifying fillers and the load-speed parameter in the tribological tests." However, in the main text, no research proof is given, only in the abstract and conclusion.

4.  In this paper, the author studies the counterpart surface Ra roughness of 0.20–0.25 µm at room temperature. Is there any speciality?

5. It is suggested to add references in the past five years in this paper to strengthen the timeliness.

6. The conclusions of this paper is too long and general.

Author Response

Esteemed reviewer! Author express greatest altitudes for attentive reading and fair evaluation of the manuscript. We completely agree with your characterization of the text and all the comments. We have followed them to revise the manuscript. Extra information has been moved to the supplementary part. We do hope that the revised version will meet your requirements.

  1. In the introduction part of this paper, it is not read coherently enough due to the lack of good logic. There should be some hierarchy and order in the questions about the background, need and current situation of the study. It is recommended to revise the logical order of the introduction.

Thank you for the comment. You are absolutely correct. We have enhanced the analysis of the state of the art and have more exactly specified the problems to be solved.

  1. According to the title of this paper, it is also important to analyze the mechanism of the effect of polymer matrices in the formation of tribological layers in line contacts.

Thank you for this relevant remark. Firstly, the title has been slightly modified since besides the counterface material, the PV product was varied as well. While revising the text, we have focus on the polymer matrix influence on the tribological layer formation. We do hope, that the information presented is enough to cast a light upon the problem under discussion.

  1. In the abstract, the authors claim: "The tribological layer formation patterns were determined by the chemical structure and flexibility of the polymer chains, the activity of the counterpart materials, the modifying fillers and the load-speed parameter in the tribological tests." However, in the main text, no research proof is given, only in the abstract and conclusion.

Esteemed reviewer! This remark continues the previous one. We do agree and have made our best to make the required corrections. Thank you for kind piece of advice and making us to focus on this issue.

  1. In this paper, the author studies the counterpart surface Raroughness of 0.20–0.25 µm at room temperature. Is there any speciality?

Esteemed reviewer! The Ra roughness of 0.20–0.25 µm is a standard value for the surface of industrial bearings. Moreover, it is very wide spread in numerous papers on metal-polymer tribological rubbing. We compared the results with ball-on-disk tests, where the counterface roughness was 10 times lower. In addition, there is nothing special in room temperature testing. However, the temperature can increase largely in the friction zone for enhancing P*V product. It was of interest to follow the effect of heating of tribological layer formation and evolution. However, your remark is of importance, and some comments have been added to the text to stress the peculiarities of this study.

  1. It is suggested to add references in the past five years in this paper to strengthen the timeliness.

Based on your first remark, the literature survey has been extended. There are a plenty of recent papers that have been added to the list of references. Thank you for this piece of advice.

  1. The conclusions of this paper is too long and general.

We completely agree. The correspondent corrections have been made, and the conclusion has been revised.

We highly appreciate all your time and effort. It has helped us to improve the manuscripts.

Reviewer 2 Report

This manuscript is well written and carefully prepared. However, the main concern of the submitted manuscript is that the paper is written in a style for the book chapter or dissertation, not suitable for a peer-review journal paper. There are too many experimental figures and tables with NO focus, and is hard to follow the idea of the authors. The explanation of each figure and table is too long and not necessary in many cases. It is therefore strongly recommended to shorten the length of the paper and concentrate on the main points of the research.

Author Response

Esteemed reviewer! Author express greatest altitudes for attentive reading and fair evaluation of the manuscript. We completely agree with your characterization of the text. We have followed your recommendations and revise the manuscript. Extra information has been moved to the supplementary part. We do hope that the revised version will meet your requirements.

Round 2

Reviewer 1 Report

The author has made detailed revisions as required and agreed to accept and publish. In addition, the reference format is further improved, such as references 19, 20, etc.